# Adequacy of Nutrients Intake among Jordanian Pregnant Women in Comparison to Dietary Reference Intakes

**DOI:** 10.3390/ijerph16183440

**Published:** 2019-09-17

**Authors:** Reema F. Tayyem, Sabika S. Allehdan, Razan M. Alatrash, Fida F. Asali, Hiba A. Bawadi

**Affiliations:** 1Department of Nutrition and Food Technology, Faculty of Agriculture, The University of Jordan, Amman 11942, Jordan; 2Department of Health Education, Jordan University Hospital, Amman 11942, Jordan; 3Department of Obstetrics and Gynaecology, Faculty of Medicine, Hashemite University, Zarqa 13115, Jordan; 4Department of Human Nutrition, college of Health Sciences; QU-health; Qatar University, Doha 2713, Qatar

**Keywords:** nutrients intake, pregnant women, dietary reference intakes

## Abstract

Objective: Maternal nutrition is considered an important pillar in the pregnancy outcomes for both mother and infant. A mother’s malnutrition and inadequate nutrient intake is associated with many undesirable pregnancy outcomes. Hence, assessing the nutritional status of the mother in the early stages of the pregnancy and preventing any inadequacy can preclude many health problems for both mother and infant. Therefore, this study aimed to assess the adequacy of nutrient intakes among Jordanian pregnant women as compared to their corresponding dietary reference intakes (DRIs). Methods: This cross-sectional study was conducted at a major University Hospital in Jordan. Three hundred pregnant women were invited to participate in the study and 286 agreed to participate. Fifty pregnant women were enrolled at week 9, then 96 pregnant women were at week 20 and 137 pregnant women were at week 30 of pregnancy. The participants completed the interview-based demographic questionnaire, pregnancy physical activity questionnaire, and quantitative food frequency questionnaire (FFQ). Results: The mean energy intake was 2768.9 ± 767.8 kcal/day and it was significantly higher in the 3rd trimester (*p* < 0.05). Women in the 3rd trimester consumed significantly more protein, carbohydrates, and sugar than women in the 1st and 2nd trimesters (*p* < 0.05). The pregnant women in the 3rd trimester consumed more sodium than women in the 1st and 2nd trimesters (*p* < 0.05). The vitamin K intake was significantly (*p* = 0.045) lower in the 2nd trimester than the 1st and 3rd trimesters. The calcium intake was significantly higher in the 3rd trimester than the 1st and 2nd trimesters (*p* = 0.021). The total micronutrient (vitamins B1, B2, B3, B6, B12, and D, calcium, and iron) intakes derived from dietary supplements and food sources throughout the 3 trimesters was significantly higher in the 3rd trimester than the 1st and 2nd trimesters (*p* < 0.05). The vitamin D, calcium, and iron intakes had the most significant increases between the 1st and 3rd trimesters (*p* < 0.001), while folic acid intake was significantly higher in the 1st trimester than the 2nd and 3rd trimester (*p* < 0.001). Most women exceeded the tolerable upper intake level (UL) for sodium in all trimesters, while 82% of women exceeded the UL of folic acid in the 1st trimester and from the supplement, not the diet. Conclusion: While the intake of some nutrients from food alone remains below the DRIs in the diets of pregnant women, the intake of other nutrients is above the UL. Raising the awareness of pregnant women about their diet and how a supplement intake can reduce the risk of inadequate intake for many micronutrients and improve their pregnancy outcomes is of great importance.

## 1. Introduction

A pregnant woman’s diet can play an important role in her health, as well as the health of her fetus and later, her child [1]. Moderate increases in energy, macronutrients and most of the micronutrient intakes in the 2nd and 3rd trimester throughout the pregnancy period are needed for healthy pregnancy outcomes. Maternal malnutrition is linked to negative pregnancy outcomes including excessive or poor maternal weight gain, increased rates of preterm birth, fetus growth retardation, and maternal and infant morbidity and mortality [2]. The suboptimal intake of macronutrients and some micronutrients is considered critical for the development and the growth of the fetus. Early nutrition is one of the basic early factors that can influence the development of the fetus and consequently, the function of all body systems, including the immune and neurological system development, encoding of the metabolism, and other developmental and physiological processes [3]. It is now well-established that folate and iron deficiency in preconception and during pregnancy may lead to many health problems which may affect the mother and fetus [4]. Therefore, assessing the nutritional status of the mother in the early stages of the pregnancy and correcting any deficiencies can hinder many health problems for both the mother and infant.

Even though the fertility rate in Jordan is relatively high, there are still many problems affecting infant outcomes including malnutrition. However, Jordan is witnessing a decline in its high fertility rates [5]. Many factors in Jordan are affecting an infant’s morbidity and mortality rate including multigravida, mother malnutrition, fluctuations in financial capabilities, the health care system, poverty, education and many other factors [6]. The appropriate maternal nutrition is one of the most effective factors that can save maternal health and life. For example, a vitamin A deficiency can be a major contributor towards nutritional anemia, maternal mortality, poor pregnancy and lactation outcomes, as well as the visual impairment of the infant in some cases. Iodine-deficient, pregnant women also face an increased risk of producing infants with disabilities [6]. The Jordan Population and Family Health Survey (JPFHS) 2002 reported that 28 per cent of women interviewed had neither bought iron supplements during their pregnancy, nor received these from health facilities. The 2004 baseline survey found that less than 40 per cent were taking vitamins, and that women’s own nutrition awareness was low [6].

Based on the aforementioned information, the present study aimed to assess nutrient intakes in a sample of Jordanian pregnant women in comparison with dietary reference intakes (DRIs). To the researchers’ knowledge, this study is the first study which assessed the nutrient intakes of pregnant women during the 3 trimesters using a validated food frequency questionnaire (FFQ) and compared them to the DRIs in Jordan.

## 2. Methods

### 2.1. Study Design and Participant

The current study is a cross-sectional observational research study designed to assess the dietary and nutritional status of women during pregnancy. The study was conducted from March 2017 to December 2018 in the Jordan University Hospital. Three hundred pregnant Jordanian women were invited to participate in this study during their antenatal visit at the maternity clinics. Two hundred and eight six out of 300 pregnant women agreed to participate in this study. Healthy Jordanian women aged 18 years or older at enrollment with a singleton pregnancy were included. Women with severe nausea and/or vomiting, gestational diabetes, preeclampsia, or any other chronic diseases that require dietary modifications such as diabetes, renal disease, liver disease, and inflammatory and autoimmune disorders were excluded. Fifty pregnant women at week 9 (weeks 9 to 13 of gestation), 96pregnant women at week 20 (weeks 14 to 26 of gestation), and 137 pregnant women at week 30 (weeks 27 to 36 of gestation) were enrolled and interviewed to complete a demographic l questionnaire, pregnancy physical activity questionnaire (PPAQ), and previously validated quantitative food frequency questionnaire (FFQ). The gestational age was determined based on the date of the last menstrual period and the ultrasound fetal biometrics by an obstetrician [7].

All participants signed an informed consent form prior to their enrollment in the study. The study was conducted according to the guidelines in the Declaration of Helsinki and the study protocol was approved by the Hashemite University Ethics Committee and Institutional Review Board of Jordan University Hospital (10/2016/3341).

### 2.2. Data Collection

#### 2.2.1. Socio-Demographic Data

The information about the maternal age, pre-pregnancy weight, education level, monthly income, and smoking status was obtained by an interviewer-administered structured questionnaire.

#### 2.2.2. Dietary Intake Assessment

A previously validated Arabic quantitative FFQ that included 117 food-items was used to assess dietary intake [8]. The FFQ was validated among a sample of 131 pregnant women using three 24-h dietary recalls. The FFQ assessed the usual dietary intake of the participants over a period of 1 month. The FFQ had a moderate reproducibility and acceptable relative validity in assessing most of the nutrient intakes among pregnant Jordanian women. The intra-class correlation coefficients ranged between −0.27 to 0.85 and energy-adjusted and de-attenuated Pearson correlation coefficients between FFQ and 24-h dietary recalls ranged from 0.01 to 0.90 [8].

A trained dietitian asked the participants how often, on average, they had consumed each food item over the past month. The questionnaire had 10 frequency selections that ranged from “never” during the past 4 weeks to “≥6/day” for beverages, and nine frequency selections that ranged from “never” during the past 4 weeks to “≥2/day” for foods. The portion sizes of each food item were similar to standard measuring units (eg, cups, tablespoons and teaspoons) or natural units (eg, 1 apple, 1 egg, a can of soft drink and 1 cracker). The participants were asked to estimate the portion size of food items in three categories: Small, medium, and large. The food models and measuring cups and spoons were used to help the participants estimate the consumed portion size of the foods.

The data concerning food preparation and cooking techniques were also collected: The use of specific types of oil, butter, and margarine; the frequency of use of dietary supplements. The FFQ included questions regarding the consumption of multiple vitamin and mineral supplements, single vitamins and minerals and herbals. The participants were asked to identify how often they consumed dietary supplements, and the brand name, dosage and duration of consumption.

To estimate the energy, the macronutrient and micronutrient intakes, and the intake of food obtained from the FFQ were analyzed using the Food Processor Nutrition analysis software (ESHA’s Food Processor SQL, version 11.6.0; Salem, OR, USA). The additional data on food consumed in Jordan was obtained from the food composition tables and added to the Food Processor Nutrition analysis software [9].

#### 2.2.3. Anthropometric Assessment

The participant’s weight and height were measured using standardized techniques and calibrated tools. The participants were weighed without shoes using a Health O meter Professional scale to the nearest 0.1 kg and height was measured without shoes to the nearest 0.1 cm using a wall mounted plastic height rod (Health O meter Professional). The pre-pregnancy body mass index (BMI) was computed and classified according to World Health Organization guidelines [10].

#### 2.2.4. Physical Activity Assessment

A semi-quantitative pregnancy physical activity questionnaire (PPAQ) was used in this study to determine the physical activity level. It was validated among a sample of 54 pregnant women using 7 days of accelerometer measurement [11]. The 1-week test–retest reproducibility has been adequate. The intraclass correlations ranged from 0.78 to 0.93 for total activity. The Spearman correlations between the PPAQ and three published cut points used to classify actigraph data varied from 0.08 to 0.43 for total activity, 0.25 to 0.34 for vigorous activity, 0.20 to 0.49 for moderate activity, and −0.08 to 0.22 for light-intensity activity. In the PPAQ, the participants were asked to recall the amount of time spent on participation in 36 types of activities grouped into four categorizations: Household/caregiving (16 activities); occupational (5 activities); sports/exercise (9 activities = 7 questions + two open questions, allowing the participants to recall any activities not previously itemized); transportation (3 activities) and inactivity (3 activities) in the current trimester. These activities were also categorized based on the intensity level (sedentary activities, light intensity activities, moderate intensity activities and vigorous-intensity activities) [11]. The physical activity levels were estimated by ranking the participants according to the total amount of time they spent in moderate and high-intensity activities (min/day). According to the Institute of Medicine guidelines for the general adult population which includes pregnant women [12], the participants were either considered sedentary (if they engaged on moderate-intensity activity for less than 30 min/day), low-active (if they engaged on moderate-intensity activity for 30–60 min/day), active (if they either engaged on moderate-intensity activity for 60 to 180 min/day or high-intensity activity for 30 to 60 min/day) or very active (if they either participated on moderate-intensity activity for more than 180 min/day or high-intensity activity for more than 60 min/day) [12].

### 2.3. Statistical Analysis

The data was analyzed using the Statistical Package for Social Sciences version 22 (IBM Corp: Armonk, NY, USA). A *p*-value <0.05 was considered statistically significant. The means and standard deviation (SD) were calculated for the continuous variables (maternal age, monthly income, gestational age, pre-pregnancy weight, height, pre-pregnancy BMI, and physical activity). The frequencies and percentages were calculated for categorical variables (pre-pregnancy BMI category, education level, and smoking status). For each trimester, the means and SD for the energy, percentage of energy from proteins, carbohydrates, fats, saturated fats, monounsaturated fats and polyunsaturated fats as well as macro-and micronutrient intakes were estimated from the FFQ. The intakes of certain micronutrients were calculated by combining the intake from supplements and the intake from food which derived from the FFQ. The Shapiro-Wilk test was used to assess the normality of the distribution of energy and nutrient intakes. A one-way analysis (ANOVA) and Fisher’s least significant difference (LSD) post hoc test were used to detect the differences across the trimesters.

The total energy and macronutrient intakes were compared with the dietary reference intakes (DRIs) by calculating the proportions of women that had intakes below or above the corresponding DRI. The energy intake was compared with the estimated energy requirements (EERs). The EERs were calculated for each trimester by using the pre-pregnancy weight, age, height, and physical activity coefficient corresponding to the physical activity level estimated by the PPAQ. An additional 340 kcal and 452 kcal were added to the 2nd and 3rd trimester EERs, respectively [12]. The protein, carbohydrate, and fat intakes as percentages of energy were compared with the acceptable macronutrient distribution range (AMDR) [13]. The protein intakes (g/day) were also compared to the estimated protein requirements (EPRs). The EPRs were calculated as 1.1 g/kg of pre-pregnancy weight for the first half of the pregnancy, to which 25g of protein per day was added for the second half of the pregnancy [12]. The carbohydrate and dietary fiber intakes were compared with the recommended dietary allowance (RDA) and adequate intake (AI).

The intakes of micronutrients from food sources alone and the combined intake (food and supplement) were compared with the DRIs by calculating the percentages of women that had intakes below the estimated average intakes (EARs) and above the upper intake limit (UL), as applicable [12].

## 3. Results

### 3.1. The Characteristics of the Participants

Table 1 presents the main characteristics of the 283 participants across pregnancy trimesters in terms of maternal age, monthly income, pre-pregnancy weight, height, pre-pregnancy BMI, smoking status, and physical activity level. The mean gestational ages of the women at enrollment was 9.0 ± 3.1, 20.0 ± 3.7, and 30.0 ± 3.2 weeks in 1st, 2nd and 3rd trimesters of pregnancy, respectively. The participants at the 1st trimester had the lowest body weight and BMI at enrollment as compared with women in the 2nd and 3rd trimesters. Based on the value of the pre-pregnancy BMI, 50%, 61.5%, and 58.1% of the participants in the 1st, the 2nd and the 3rd trimesters of pregnancy had a normal weight; 6.3%, 5.2%, and 2.9% were underweight; 33.3 %, 28.1%, and 26.5% were overweight, and 10.4%, 5.2%, and 12.5% were obese. A large proportion of the participants, 68%, 79.2%, and 63.6%, had a college degree and above in the 1st, the 2nd and the 3rd trimesters of pregnancy, respectively.

### 3.2. Macronutrients Intake

Table 2 shows the mean ± SD of the energy and macronutrient intakes of all the participants derived from the FFQ across the three trimesters of pregnancy. The mean energy intake was 2768.9 ± 767.8 kcal/day with a wide range of variability and a significantly higher intake in the 3rd trimester (*p* < 0.05). No significant difference was observed for the percentage of energy from proteins, carbohydrates, and saturated fats. However, a significant increase in fats, monounsaturated fats, and polyunsaturated fats as percentages of the energy intake was observed among pregnant women in the1st trimester as compared to women in their 2nd and 3rd trimesters. The pregnant women consumed significantly more protein, carbohydrates, and sugar in the 3rd trimester than the 1st and 2nd trimester (*p* < 0.05). This increase in intakes of protein and carbohydrates explains the increase in calories among women in their third trimester. The proportions of women that reported energy and macronutrient intakes as well as protein, carbohydrate, and fat intakes as percentages of energy above or below the corresponding DRIs are shown in Table 3. The mean energy intake exceeded the estimated energy requirements (EERs) in all trimesters. However, half of the women had energy intakes below the EERs. The protein intake as a percentage of energy for most participants was within the acceptable distribution range (10–35%) in all trimesters. Approximately 50 % of the participants had protein intakes below the estimated protein requirements (EPRs), but the mean protein intake exceeded the estimated requirements in all trimesters. Similarly, carbohydrate intake as a percentage of energy for the majority of women was within the acceptable distribution range (45–65%). For all participants (100%), the carbohydrate intake was above RDA (175 g/day) in each trimester. More than 50% of participants had dietary fiber intake below the AI of 28 g/day in all trimesters. On the other hand, 60%, 54.2, and 37.2 % of women in the 1st, the 2nd and the 3rd trimesters of pregnancy reported fat intakes that were above the acceptable distribution range as a percentage of the energy intake, respectively.

### 3.3. Micronutrients Intake Obtained from Food Sources Only

The micronutrient intakes derived from the food sources only for the participants in the 1st, 2nd, and 3rd trimesters are presented in Table 4. The pregnant women consumed significantly more sodium in the 3rd trimester than the 1st and 2nd trimesters (*p* = 0.019). On the contrary, significantly less vitamin K (*p* = 0.045) in the 2nd trimester than the 1st and 3rd trimesters were consumed. The calcium intake was significantly higher in the 3rd trimester than the 1st and 2nd trimesters (*p* = 0.021). The increase in micronutrient consumption among women in the 3rd trimester may be attributed to their increasing intakes of energy, carbohydrates and protein (Table 1). A high prevalence of inadequate intake was found for vitamin D (100.0%, 99.0%, 100.0%), vitamin E (92.0%, 99.0%, 95.6%), vitamin B6 (84.0%, 89.6%, 83.2%), iron (96.0%, 96.9%, 92.0%), iodine (98.0%, 97.1%, 99.3%), magnesium (72.0%, 66.7%, 70.8%), and zinc (90%, 87.5%, 92.0%) in all trimesters, when only food sources were considered (Table 5). Vitamins A and B12 intakes were below the EAR for 58.0%, 60.4%, and 47.4%and 66.0%, 63.5%, and 46.7% of women in the 1st, 2nd, and 3rd trimesters, respectively. Approximately one third of women that participated in the study showed inadequate intakes of thiamin, riboflavin, niacin, vitamin C, calcium, copper, phosphorus and selenium across pregnancy. Folate intake was inadequate for 45.8% of the participants in the 2nd trimester but only for 28.0% and 34.3% of women in the 1st and 3rd trimesters, respectively. In all trimesters, most of the participants reported sodium intake above the UL of 2300 mg, while the mean intake of the potassium was below the AI (Table 5).

### 3.4. Micronutrients Intake Obtained from Dietary Supplements and Food Sources

Table 6 describes total micronutrient intakes derived from combining dietary supplements and food sources trough the trimesters of pregnancy. Vitamins B1, B2, B3, B6, B6, B12, and D, calcium, and iron intakes were significantly higher in the 3rd trimester than the 1st and 2nd trimesters (*p* < 0.05). The iodine and zinc intakes were significantly higher in the 2nd and 3rd trimesters than the 1st trimester (*p* < 0.05). Vitamin D, calcium, and iron intakes had the most significant differences between the 1st and 3rd trimesters (*p* < 0.001). On the contrary, folic acid intake was significantly higher in the 1st trimester than the 2nd and 3rd trimesters (*p* < 0.001).

As shown in Table 7, when food sources and dietary supplements were combined, the proportion of the participants with adequate micronutrient intakes increased, particularly in the 2nd and 3rd trimesters, with the exception of vitamin B6, vitamin B12, vitamin E, iodine, magnesium, and zinc. The total intake of folic acid was above the UL for a majority of women at the 1st trimester. The total vitamin D intake was below EAR for a majority and half of women in the 1st and 2nd trimesters, respectively, whereas 21.2 % of women in the 3rd trimester had inadequate intakes of vitamin D. The total iron intake was inadequate for 90% of the participants in the 1st trimester but only for 25.0% and 5.8 % of women in the 2nd and 3rd trimesters, respectively (Table 7).

## 4. Discussion

This study aimed to assess adequacy of nutrient intakes among Jordanian pregnant women. Our study findings revealed that a large proportion of women may not be meeting their nutritional requirements during pregnancy from diet alone. Approximately 50% of the pregnant women in our study showed that the intake of total energy and protein was below the AMDR during pregnancy, while fiber intake was below the AMDR in approximately 74% of the women. However, protein and carbohydrate intakes, as percentages of total energy, were in the acceptable distribution range for about 98% of pregnant women. Most of our findings are in agreement with Liu et al. who stated that most pregnant women of their study had imbalanced macronutrient distribution and excessive energy intake which derived mainly from fat [14]. Additionally, Dubois et al. documented that one third of their study participants had total fat intakes that exceeded the AMDR [15]. The protein and carbohydrate intakes, as percentages of total energy, were in the acceptable distribution range for a majority of pregnant women. They also reported a median intake for dietary fiber below the recommended level AI, which is consistent with our study findings. This could be attributed to the high intake of fat at the expense of carbohydrate and protein [15].

The majority of pregnant women had usual intakes below the EARs, especially for iron, iodine, zinc, vitamins D and E in all trimesters. The dietary intakes of vitamins B2, B3 and C, calcium, phosphorus, and selenium were found to below the EARs in approximately 7–27% of women during the 3rd trimesters. Whereas the mean intake of potassium seemed to be low relative to the AI for most women and sodium intake was above the UL. The higher intake of sodium in most pregnant women globally could be attributed to the excessive consumption of salted and processed foods. It is also expected that from the high sodium intake, there is low potassium consumption.

Some of our findings are in agreement with Liu et al. who revealed that the intakes of vitamins A and B6, calcium, magnesium, and selenium were all below Chinese Recommended Nutrient Intake (RNI) and EARs in all trimesters [14]. Furthermore, the findings of Dubois et al. showed that the majority of Canadian pregnant women (85%) had sodium intakes above the UL. The median intakes for fiber and potassium were lower than AI. The dietary intakes of vitamin B6, magnesium, and zinc were below the EARs for 10–15% of the women. The majority of the women had dietary intakes below the EARs for iron (97%), vitamin D (96%), and folate (70%) [15].

The inadequate consumption of iodine among almost all the (>97%) participants at all trimesters has been detected which may be due to the low intake of iodine dietary sources such as fish, sea foods, and iodized salt. Moreover, Jordan is considered a mountainous country and far from the sea which makes the ingestion of iodine more likely to be low and inadequate. The findings of our study are also consistent with many findings that have been reported by Lee et al. who compared dietary intakes of women during pregnancy in low- and middle-income countries (Caribbean and Central/South America than in Africa and Asia) to the DRIs [16]. The authors revealed that Jordan is above the medians of energy, fat, protein and vitamin C intakes as compared to the other countries included in their review, while carbohydrate, calcium and vitamin A intakes were near to the median of the other countries’ intake [16]. Conversely, Lee and his colleagues found that pregnant women in Jordan showed a lower intake (below the median) in folate, iron and zinc as compared to the countries included in the study. Their results were based on diet intake only and supplements were not included [16].

When combining the nutrients from food sources with dietary supplements, micronutrient intakes (thiamine, niacin, vitamins B6, B12, and D, calcium, and iron) were significantly higher among women in their 3rd trimester as compared to women in their 1st and 2nd trimesters (*p* < 0.05). Vitamin D, calcium, and iron intakes showed the most significant differences between the 1st and 3rd trimesters (*p* < 0.001). The findings of this study showed that the proportion of the participants with adequate micronutrient intakes increased particularly in the 2nd and 3rd trimesters except for vitamins B6, B12 and E, iodine, magnesium, and zinc. Another study by Bailey et al. suggests that a significant number of pregnant women are not meeting the recommendations for vitamins D, C, A, B6, K, and E, as well as folate, iron, calcium, potassium, magnesium, and zinc, even with the use of dietary supplements [17].

Another finding involves the higher intake of folic acid by women in the 1st trimester in comparison with women in the 2nd and 3rd trimesters (*p* < 0.001). This could be due to the emphasis of folic acid supplement intake during the 1st trimester. The total intake of folic acid was above the UL for most of women at the 1st trimester. Although folic acid supplementation is very essential to prevent neural tube defects, excessive folic acid intake to dosages exceeding the UL (≥1000 μg/day) during the periconception period has been associated with lower levels of cognitive development in children aged 4–5 y as demonstrated by Valera-Gran et al. [18]. Therefore, using folic acid supplementation dosages ≥1000 μg/day during pregnancy should be monitored and prevented as much as possible, unless medically prescribed as recommended by the authors [18].

The inadequate intake of vitamin D from diet alone was below EAR in nearly all our study participants and the diet and supplement were below EAR for a majority and a half of the women in the 1st and 2nd trimesters, respectively, and 21.2% of women in the 3rd trimester. Similar results were reported by Al-Faris [19]. Al-Faris revealed that adequate vitamin D intake (≥600 IU/day) was reported among only 8.1% of Saudi pregnant women [19]. The author attributed that to most foods being consumed by her sample not containing significant amounts of vitamin D to achieve vitamin adequacy [19]. In Saudi Arabia, the lifestyles and dietary patterns have transitioned remarkably toward western diets and sedentary lifestyles focusing mainly on indoor activities and energy-dense foods, such as fast food. All of this causes a reduction in vitamin D intake levels [19].

The total iron intake was inadequate for 90% of the participants in the 1st trimester but only for 25.0% and 5.8 % of women in the 2nd and 3rd trimesters, respectively. Dubois et al. stated that the prevalence of inadequate intake was 10% for all nutrients except vitamin D (18%) and iron (15%) when the micronutrient intakes were considered from both the food and supplements, whereas 32% and 87% of the women had total intakes above the ULs for iron and folic acid, respectively [15]. Most of the pregnant women were at risk of excessive consumption of folic acid and iron when supplements were taken [17].

Another study by Mosha et al. (2017) showed that 99% (n 7602) of Tanzanian pregnant women had a total daily dietary iron intake below the RDA and 90·1% (n 6880) of women had daily dietary calcium intake below 1200 mg [20].

The main strengths of this study were the use of a FFQ containing foods culturally adapted to Jordanian pregnant women. A trained nutritionist conducted in-person interviews to collect all the required data and to minimize missing data. The food models and measuring tools were used to estimate portion sizes. However, the present study had many limitations. The main limitation was the fact that FFQ had sources of errors including recall bias and overestimating or underestimating of nutrient intakes. Like all other studies, it was difficult to take into consideration the possible effects of cooking on the bioavailability of the various nutrients. In addition, using the cross-sectional design might not reflect the real increase in nutrient intakes throughout the 3 trimesters of the pregnancy.

In conclusion, the intake of some nutrients from food alone was below the DRIs in the diets of pregnant women. However, supplement use reduces the risk of inadequate intake for many micronutrients. Our study findings highlight the importance of raising the awareness of pregnant women and educating them about their diet in order to improve their pregnancy outcomes.

## Figures and Tables

**Table 1 ijerph-16-03440-t001:** Sociodemographic characteristics of participants (*n* = 283) enrolled in a cross-section study to assess nutrient intakes of Jordanian pregnant women.

Variables	Pregnancy Trimesters		
	First (*n* = 50)	Second (*n* = 96)	Third (*n* = 137)
	Mean ± SD		
Maternal age (year)	29.7 ± 5.2	30.0 ± 5.2	29.3 ± 5.1
Monthly income (Jordanian Dinars)	612.1 ± 300.2	637.4 ± 312.4	557.3 ± 350.1
Gestational weeks	9.0 ± 3.1	20.0 ± 3.7	30.0 ± 3.2
Body weight (kg) at enrollment	66.4 ± 12.5	68.2 ± 12.0	75.7 ± 13.3
Pre-pregnancy weight (kg)	65.1 ± 13.7	63.1 ± 11.1	64.1 ± 10.6
Height (m)	1.6 ± 0.07	1.6 ± 0.06	1.6 ± 0.05
Pre-pregnancy BMI (kg/m^2^)	24.9 ± 4.6	24.1 ± 3.9	24.5 ± 3.8
BMI (kg/m²) at enrollment	24.7.0 ± 4.1	26.1.0 ± 4.4	28.1.0 ± 3.7
Physical activity level (minutes of moderate and vigorous activity/day)	58.9 ± 10.8	67.7 ± 8.5	56.3 ± 4.7
***n* (%)**			
**Pre-pregnancy BMI category**			
Under weight (<18.5 kg/m^²^)	3.0 (6.3%)	5.0 (5.2%)	4.0 (2.9%)
Normal (18.5–24.9 kg/m²)	24.0 (50%)	59.0 (61.5%)	79.0 (58.1%)
Overweight (25–29.9 kg/m²)	16.0 (33.3%)	27.0 (28.1%)	36.0 (26.5%)
Obese (>30 kg/m²)	5.0 (10.4%)	5.0 (5.2%)	17.0 (12.5%)
**Education level**			
Illiterate	0 (0%)	1.0 (1%)	1.0 (0.7%)
Primary	5.0 (10%)	3.0 (3.1%)	14.0 (10.2%)
High school Degree	11.0 (22.0%)	16.0 (16.7%)	35.0 (25.5%)
Diploma and above	34.0 (68%)	76.0 (79.2%)	87.0 (63.6%)
Smoking during pregnancy	1.0 (2%)	1.0 (1%)	0 (0%)

*n*: number of participants; SD: standard deviation. Body mass index (BMI) categories according to WHO classification [10].

**Table 2 ijerph-16-03440-t002:** The mean daily intakes of energy and macronutrients estimated by a validated food frequency questionnaire administered to 283 pregnant Jordanian women.

Energy and Macronutrients	All Participants (*n* = 283)	First Trimester (*n* = 50)	Second Trimester (*n* = 96)	Third Trimester (*n* = 137)	*p*-Value
	Mean ± SD				
Energy (kcal/day)	2768.9 ± 767.8	2611.5 ± 580.0 ^a^	2632.6 ± 606.1 ^a^	2922.0 ± 893.7 ^b^	0.015
Energy from proteins (%)	13.7 ± 1.7	13.7 ± 1.8	13.5 ± 1.8	13.7 ± 1.6	0.670
Energy from carbohydrates (%)	53.7 ± 5.1	52.8 ± 5.1	53.4 ± 5.4	54.3 ± 4.8	0.119
Energy from fats (%)	34.6 ± 4.8	35.5 ± 4.4 ^b^	35.2 ± 5.2 ^ab^	33.8 ± 4.5 ^a^	0.044
Energy from saturated fats (%)	10.2 ± 2.2	10.7 ± 2.6	10.40 ± 2.2	9.9 ± 2.1	0.069
Energy from monounsaturated fats (%)	8.3 ± 3.1	9.2 ± 3.0 ^b^	8.8± 3.0 ^ab^	7.7 ± 3.1 ^a^	0.002
Energy from polyunsaturated fats (%)	4.5 ± 1.7	5.0 ± 2.1 ^b^	4.7 ± 1.7 ^ab^	4.2 ± 1.5 ^a^	0.020
Proteins (g/day)	94.3 ± 25.9	89.2 ± 21.4 ^a^	89.3 ± 22.8 ^a^	99.6 ± 24.5 ^b^	0.009
Total Carbohydrates (g/day)	371.4 ± 108.6	343.8 ± 79.0 ^a^	350.8 ± 89.0 ^a^	396.0 ± 24.5 ^b^	0.002
Dietary Fiber (g/day)	26.9 ± 9.3	25.9 ± 8.0	25.1 ± 7.3	28.5 ± 10.7	0.069
Sugar (g/day)	137.460.6	125.3 ± 44.8 ^a^	128.0 ±54.4 ^a^	148.4 ± 67.7 ^b^	0.043
Fats (g/day)	107.9 ± 35.2	104.4 ± 27.9	104.0 ± 28.6	111.8 ± 41.2	0.431
Saturated Fats (g/day)	31.5 ± 11.2	31.1 ± 9.6	30.6±10.1	32.3 ± 12.5	0.733
Monounsaturated Fats (g/day)	25.4 ± 12.1	26.4 ± 10.0	25.6 ± 9.7	24.9 ± 14.1	0.338
Polyunsaturated Fats (g/day)	13.7 ± 6.2	14.4 ± 6.3	13.5 ± 5.0	13.6 ± 7.0	0.668
Trans-Fats (g/day)	0.95 ± 0.12	0.7 ± 0.20	1.2 ± 0.4	0.72 ± 0.24	0.694
Cholesterol (mg/day)	208.5 ± 95.8	199.1 ± 80.7	199.4 ± 99.6	218.2 ± 97.8	0.164
Omega-3 Fatty Acids (g/day)	0.73 ± 0.31	0.70 ± 0.22	0.74 ± 0.23	0.71 ± 0.34	0.773
Omega-6 Fatty Acids (g/day)	9.2 ± 5.1	10.2 ± 5.4	9.3 ± 4.4	8.7± 5.5	0.159

SD: Standard deviation. One-way ANOVA and Fisher’s LSD post hoc test were used. Data were considered statistically significant at *p* < 0.05. Means within the same row with different superscript letters are significantly different.

**Table 3 ijerph-16-03440-t003:** The proportions of pregnant women that reported energy and the micronutrient intakes and protein, carbohydrate, and fat intakes as percentages of energy below or above the corresponding dietary reference intakes.

	First Trimester (*n* = 50)			Second Trimester (*n* = 96)			Third Trimester (*n* = 137)		
Energy and Macronutrients	DRI	% Below DRI	% Above DRI	DRI	% Below DRI	% Above DRI	DRI	% Below DRI	% Above DRI
EER (kcal/day) *	2004.6 ± 171.4	54.0	46.0	2251.1 ± 126.2	54.7	45.3	2352.8 ± 123.2	52.8	48.2
Energy from proteins (%)	10–35	2.0	0	10–35	3.2	0	10–35	0	0
EPR (g/day) *	71.6 ± 15.1	56.0	44.0	94.4 ± 12.2	56.3	43.7	95.6 ± 11.6	48.2	51.8
Energy from carbohydrates (%)	45–65	2.0	2.0	45–65	4.2	2.1	45–65	3.6	1.5
Carbohydrates	175	0	100	175	0	100	175	0	100.0
Dietary Fiber (g/day)	28	74.0	26.0	28	69.8	30.2	28	54.0	46.0
Energy from fats (%)	20–35	0	60.0	20–35	0	54.2	20–35	0	37.2

Abbreviation: SD: standard deviation; DRI: dietary reference intake; EER: estimated energy requirement. EER calculated with the following formula: 354 − (6.91 × age) + physical activity coefficient × (9.36 × pre-pregnancy weight (kg)) + (726 × height (m)), to which an additional 340 or 452 kcal were added in the second and third trimesters, receptively [12]. EPR: estimated protein requirement (1.1 g/kg or pre-pregnancy weight for the first half of pregnancy and 1.1 g/kg of pre-pregnancy weight + 25 g for the second half) [12]. * Data were represented as Mean ± SD.

**Table 4 ijerph-16-03440-t004:** The mean daily intake of total micronutrient intakes from food alone estimated by a validated food frequency questionnaire administered to 283 pregnant Jordanian women.

Micronutrients	All Participants (*n* = 283)	First Trimester (*n* = 50)	Second Trimester (*n* = 96)	Third Trimester (*n* = 137)	*p*-Value
	Mean ± SD				
Vitamin A-RAE (μg/day)	637.2 ± 23.8	687.8 ± 65.0	573.4 ± 36.7	663.4 ± 34.3	0.230
Thiamin (mg/day)	1.9 ± 0.66	1.8 ± 0.57	1.8 ± 0.63	2.0 ± 0.71	0.221
Riboflavin (mg/day)	2.0 ± 0.69	1.9 ± 0.56	1.9 ± 0.65	2.1 ± 0.75	0.066
Niacin (mg/day)	18.1 ± 5.6	17.4 ± 4.8	17.3 ± 5.1	19.0 ± 6.1	0.053
Niacin, NE (mg/day)	25.7 ± 8.0	25.1 ± 6.9	24.6 ± 7.7	26.7 ± 8.5	0.223
Vitamin B6 (mg/day)	1.1 ± 0.44	1.2 ± 0.47	1.1 ± 0.40	1.1 ± 0.46	0.402
Vitamin B12 (μg/day)	2.4 ± 1.5	2.2 ± 1.4	2.2 ± 1.3	2.7 ± 1.7	0.054
Vitamin C (mg/day)	191.8 ± 120.8	190.1±121.1	173.1 ± 108.5	205.5 ± 127.8	0.182
Vitamin D (IU/day)	68.0 ± 4.0	77.7 ± 11.4	67.7 ± 6.8	64.7 ± 5.3	0.772
Vitamin E (α-Tocopherol) (mg/day)	6.7 ± 2.9	7.3 ± 3.1	6.2 ± 2.0	6.8 ± 3.1	0.078
Folate DFE (μg/day)	579.2 ± 220.7	586.4 ± 187.6	554.4 ± 214.2	594.0 ± 235.7	0.446
Folic acid(μg/day)	212.0 ±92.2	199.6 ± 84.4	203.2 ± 89.8	222.7 ± 96.0	0.412
Vitamin K(μg/day)	182.0 ±130.5	196.0 ± 113.9 ^b^	154.4 ± 107.6 ^a^	196.2 ± 147.4 ^b^	0.045
Calcium (mg/day)	1072.8 ± 443.2	1047.6 ± 428.9 ^ab^	981.3 ± 405.6 ^a^	1146.2 ± 463.1 ^b^	0.021
Copper (mg/day)	1.1± 0.48	1.1 ± 0.46	1.1 ± 0.45	1.0 ± 0.51	0.462
Iodine (μg/day)	66.0 ± 46.4	61.1± 54.1	62.8 ± 46.4	70.1± 43.3	0.207
Iron (mg/day)	14.9 ± 4.3	14.1 ± 3.9	14.4 ± 3.7	15.4 ± 4.3	0.211
Magnesium (mg/day)	257.9 ± 95.9	265.8 ± 93.8	255.9 ± 87.0	256.4 ± 102.9	0.701
Phosphorus (mg/day)	882.3 ± 321.4	881.2 ± 309.5	852.5 ± 307.3	903.6 ± 335.6	0.525
Potassium (mg/day)	2882.4 ± 1035.6	2930.8 ± 941.5	2730.8 ± 915.6	2970.9 ± 1137.6	0.296
Selenium (μg/day)	74.9 ± 30.4	74.2 ± 28.5	71.9 ± 32.9	77.2 ± 30.4	0.300
Sodium (mg/day)	3716.5 ± 1297.5	3608.9 ± 1258.6 ^ab^	3425.7 ± 998.6 ^a^	3959.6 ± 1449.2 ^b^	0.019
Zinc (mg/day)	6.6 ± 2.3	6.5 ± 2.2	6.6 ± 2.3	6.7 ± 2.4	0.896

Abbreviations: NE: niacin equivalent, SD: Standard deviation. One-way ANOVA and Fisher’s LSD post hoc test were used. Data were considered statistically significant at *p* < 0.05. Means within the same row with different superscript letters are significantly different.

**Table 5 ijerph-16-03440-t005:** The proportions of pregnant women that reported the micronutrient intakes from food alone below or above the corresponding dietary reference intakes.

Micronutrients	EAR	UL	First Trimester		Second Trimester		Third Trimester	
			% Below EAR	% Above UL	% Below EAR	% Above UL	% Below EAR	% Above UL
Vitamin A-RAE (μg/day)	550	3000	58.0	0	60.4	0	47.4	0
Thiamin (mg/day)	1.2	-	22.0	-	21.9	-	24.1	-
Riboflavin (mg/day)	1.2	-	12.0	-	20.8	-	13.1	-
Niacin (mg/day)	-	-	-	-	-	-	-	-
Niacin, NE (mg/day)	14	35	8.0	10	5.2	10.4	7.3	13.9
Vitamin B6 (mg/day)	1.6	100	84.0	0	89.6	0	83.2	0
Vitamin B12 (μg/day)	2.2	-	66.0	-	63.5	-	46.7	-
Vitamin C (mg/day)	70	2000	6.0	0	17.7	0	10.9	0
Vitamin D (IU/day)	400	4000	100.0	0	99.0	0	100.0	0
Vitamin E (α-Tocopherol) (mg/day)	12	1000	92.0	0	99.0	0	95.6	0
Folate DFE (μg/day)	520	-	28.0	-	45.8	-	34.3	-
Folic acid(μg/day)	-	1000	-	0	-	0	-	0
Vitamin K(μg/day)	-	-	-	-	-	-	-	-
Calcium (mg/day)	800	2500	32.0	0	24.8	0	27.0	0.73
Copper (mg/day)	0.8	10	28.0	0	33.3	0	34.3	0
Iodine (μg/day)	160	1100	98.0	0	97.1	0	99.3	0
Iron (mg/day)	22	45	96.0	0	96.9	0	92.0	0
Magnesium (mg/day)	290–300	350	72.0	18.0	66.7	15.6	70.8	13.1
Phosphorus (mg/day)	580	3500	16.0	0	21.9	0	16.8	0
Potassium (mg/day)	-	-	-	-	-	-	-	-
Selenium (μg/day)	49	400	20.0	0	30.2	0	25.5	0
Sodium (mg/day)	-	2300	-	88.0	-	86.5	-	92.0
Zinc (mg/day)	9.5	40	90.0	0	87.5	0	92.0	0

Abbreviations: DFE: dietary folate equivalent; RAE: retinol activity equivalents; NE: niacin equivalent. Abbreviations: EAR: Estimated Average Requirement; UL: upper intake limit; NE: niacin equivalent; UL: upper intake limit. When no EAR or UL was established for a nutrient, the “-” is used rater than 0.

**Table 6 ijerph-16-03440-t006:** The mean daily intake of total micronutrients (including food sources and supplements) obtained from a validated food frequency questionnaire administered to 283 pregnant Jordanian women.

Micronutrients	All Participants (*n* = 283)	First Trimester (*n* = 50)	Second Trimester (*n* = 96)	Third Trimester (*n* = 137)	*p*-Value
	Mean ± SD				
Thiamin (mg/day)	2.1 ± 0.87	1.8 ± 0.57 ^a^	2.1± 0.63 ^ab^	2.2 ± 0.92 ^b^	0.043
Riboflavin (mg/day)	2.2 ± 0.90	1.9 ± 0.08 ^a^	2.1 ± 0.90 ^ab^	2.4 ± 0.95 ^b^	0.015
Niacin, NE (mg/day)	28.8 ± 10.9	25.1 ± 6.9 ^a^	28.4 ± 11.4 ^ab^	30.5 ± 11.5 ^b^	0.041
Vitamin B6 (mg/day)	1.5 ± 0.86	1.2 ± 0.47 ^a^	1.4 ± 0.92 ^ab^	1.5 ± 0.92 ^b^	0.027
Vitamin B12 (μg /day)	2.9 ± 1.9	2.2 ± 1.4 ^a^	2.8 ± 1.8 ^ab^	3.2 ± 2.0 ^b^	0.004
Vitamin C (mg/day)	203.9 ± 123.4	190.1±121.1	187.7 ± 111.7	220.3 ± 130.5	0.152
Vitamin D (IU/day)	1381.9 ±70.8	213.7 ± 74.6 ^a^	1213.0 ± 121.7 ^b^	1926.6 ± 86.5 ^c^	<0.001
Vitamin E (α-Tocopherol) (mg/day)	6.7 ± 2.8	7.3 ± 3.1	9.3 ± 6.5	10.0 ± 7.1	0.163
Folic acid (μg/day)	1010.6 ±103.3	4327.6 ± 266.9 ^b^	286.6 ± 191.7 ^a^	307.3 ± 197.6 ^a^	<0.001
Vitamin K (μg/day)	194.1 ±133.8	196.0 ± 113.9	169.0 ± 114.7	211.0 ± 150.1	0.103
Calcium (mg/day)	1403.7± 513.8	1067.6 ± 453.3 ^a^	1305.7 ± 489.6 ^b^	1595.1 ± 469.9 ^c^	<0.001
Copper (mg/day)	1.2 ± 0.62	1.1 ± 0.46	1.3 ± 0.63	1.3 ± 0.65	0.610
Iodine (μg/day)	90.3 ± 76.2	61.1± 54.1 ^a^	91.9 ± 81.6 ^b^	99.7± 77.1 ^b^	0.016
Iron (mg/day)	41.5 ± 19.5	16.9 ± 11.1 ^a^	41.0± 18.3^b^	50.8 ±14.0 ^c^	<0.001
Magnesium (mg/day)	283.9 ± 114.0	265.8 ± 93.8	278.2 ± 113.2	288.2 ± 121.2	0.465
Zinc (mg/day)	8.5 ± 5.0	6.5 ± 2.2 ^a^	8.9 ± 5.4 ^b^	9.0 ± 5.2 ^b^	0.036

Abbreviations: NE: niacin equivalent, SD: Standard deviation. One-way ANOVA and Fisher’s LSD post hoc test were used. Data were considered statistically significant at *p* < 0.05. Means within the same row with different superscript letters are significantly different.

**Table 7 ijerph-16-03440-t007:** The proportions of pregnant women that reported micronutrient intakes from food and supplements below or above the corresponding dietary reference intakes.

Micronutrients	EAR	UL	First Trimester		Second Trimester		Third Trimester	
			% Below EAR	% Above UL	% Below EAR	% Above UL	% Below EAR	% Above UL
Thiamin (mg/day)	1.2	-	22.0	-	15.6	-	20.4	-
Riboflavin (mg/day)	1.2	-	12.0	-	16.7	-	13.1	-
Niacin, NE (mg/day)	14	35	8.0	10.0	4.2	21.9	7.3	30.7
Vitamin B6 (mg/day)	1.6	100	84.0	0	71.9	0	65.0	0
Vitamin B12 (μg/day)	2.2	-	66.0	-	52.1	-	39.4	-
Vitamin C (mg/day)	70	2000	6.0	0	15.6	0	10.2	0
Vitamin D (IU/day)	400	4000	92.0	0	53.1	0	21.2	0
Vitamin E (α-Tocopherol) (mg/day)	12	1000	92.0	0	78.1	0	75.9	0
Folic acid (μg/day)	-	1000	-	82.0	-	0	-	0
Vitamin K (μg/day)	-	-	-	-	-	-	-	-
Calcium (mg/day)	800	2500	30.0	0	15.6	1.0	1.5	3.6
Copper (mg/day)	0.8	10	28.0	0	27.1	0	29.2	0
Iodine (μg/day)	160	1100	98.0	0	78.1	0	79.6	0
Iron (mg/day)	22	45	90.0	0	25.0	0	5.8	0
Magnesium (mg/day)	290–300	350	72.0	18.0	56.3	31.3	55.5	29.2
Zinc (mg/day)	9.5	40	90.0	0	68.8	0	72.3	0

Abbreviations: DFE: dietary folate equivalent; EAR: Estimated Average Requirement; RAE: retinol activity equivalents; NE: niacin equivalent; UL: upper intake limit. When no EAR or UL was established for a nutrient, the “-” is used rather than 0.

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
