# Peer review of "Adequacy of Nutrients Intake among Jordanian Pregnant Women in Comparison to Dietary Reference Intakes"

_ijerph, 2019, doi:10.3390/ijerph16183440_

Round 1
Reviewer 1 Report
Inadequate maternal nutrition during pregnancy might lead to adverse maternal and infant outcomes. Therefore, it is very important to assess nutrition during pregnancy and correct deficiencies. The authors aimed to assess nutrients intake throughout pregnancy in Jordanian women. Data were collected from demographic, physical activity and food intake questionnaires completed by 286 pregnant women throughout the 3 trimesters of pregnancy. Differences in energy and nutrient intake were found between the trimesters. The intake of some micronutrients from food only was below the DRIs and supplement use increased their level above the DRIs. However, some nutrients remained below DRIs even after taking supplements. These results point to the need of taking supplements in order to obtain adequate levels of some of the micronutrients. This study is the first in Jordan which assessed nutrients intake of pregnant women during the 3 trimesters using a validated food frequency questionnaire and compared them to the DRIs.
Please see my comments below.
Abstract
Line a3: a sentence is needed to describe the consequences of inadequate nutrition during pregnancy and the importance of assessing nutrition during pregnancy. Line 14: DRI abbreviation should be explained Lines 21-22: the differences are between different women in the 3 trimesters, therefore is more suitable to write the sentence as follows: “Women in the 3rd trimester consumed significantly more protein….than women in the 1st and 2nd trimesters.” Lines 22-23: the same as in comment 3. Line 26: “Findings of the study showed that” is not needed. Line 28: replace “through” with “throughout”. Line 30: replace differences with increases. Line 32: “trimesters” instead “trimester”. Line 33: “exceeded” instead “exceeding”. Line 33: “from the supplement not diet” should be rephrased. Much more focus should be given to the description of nutrients that were below or above the DRIs rather than the differences in intake between the trimesters. The conclusion of the abstract states this but it is not supported by the findings presented in the results section. The purpose and the meaning of comparing the levels of intake between trimesters is unclear. Which information do these comparisons provide in addition to the information whether the level is adequate or not in comparisons to the DRIs? Please add a sentence in the conclusions. The conclusions refer to the findings that some of the nutrients were below DRIs. However, other nutrients such as sodium exceeded DRIs. This also should be addressed here and in the discussion, including suggesting ways to resolve this problem.Introduction
Lines 53-56: the purpose of comparing to other countries is unclear. Lines 53-59: the sentences here should be rephrased to state that even though the fertility rate in Jordan is relatively high there are still many problems affecting infant outcomes including malnutrition.Methods
Line 98: “The FFQ was… validated…” “previously” should be added. Line 129: “previously” should be added. Lines 159-161: the cross-sectional design of the study in which different women are assessed in different trimesters is not suitable here because it doesn’t necessarily mean that all women showed increase in intake throughout pregnancy to the same extent. It would have been more appropriate to have a longitudinal study design, in which the same women assessed throughout all trimesters and use repeated measure ANOVA to analyse the within group differences across the 3 timepoints. Please address this and explain why you chose this design, including brief description of practical reasons if there were any.Results
Lines 216-221: this legend should be moved under table 3 and another one should be generated for table 2. P value should be explained that it is for the one-way ANOVA. Post-hoc test significance a and b should be explained. Line 220: replace “or” with “of”. Table 3 columns 2, 5, 8: it is unclear whether theses are the DRIs or the sample mean. It is better to split these into two columns for each trimester. Lines 231-232: this sentence belongs to the discussion and should be clarified. Line 246: legend for table 4 should be added, including an explanation of P values and post-hoc tests (a and b).Discussion
Line 282: add the name of the authors before the citation. Do the same throughout the discussion where appropriate. Lines 291-292: “… most pregnant women in our study took more folic acid in the first trimester with a significant decrease in the 2nd and 3rd trimesters” this sentence should be rephrased “women in the 1st trimester took more folic acid in comparison with women in the 2nd and 3rd trimesters. Line 292: a sentence should be added here that the level of folic acid exceeded the DRIs and the implications of this should be discussed, including whether there are negative consequences on pregnancy and infant outcomes. This is an example of a case where awareness to the importance of taking supplements exists but leads to over-consumption and therefore better education around this should be provided. Lines 290-307: the description should be either in relation to the comparison methods, i.e. describing all nutrients (or group of nutrients) that their levels found to be different across trimesters and followed by all nutrients compared to their DRIs, or describing for each nutrient the difference across trimesters and in comparison with DRIs. The way it is currently described is confusing and therefore hard to follow. Line 290 and 297: the beginning of both sentences is very similar. It should refer from the beginning to the specific nutrient. The manuscript is mainly focused on the need to take supplements to correct deficiencies in some micronutrients. However, attention should also be given to ways to reduce energy and nutrients intake that exceeded DRIs, such as sodium, increase awareness to not overly consume supplements for nutrients that exceeded DRIs, such as folic acids, and encourage supplement consumption for nutrients that were below DRIs even after taking supplements. It will be good to have a few sentences describing the literature concerning awareness and education in Jordan around maternal nutrition during pregnancy. Line 350: add another limitation about the cross-sectional design of the study for comparing intake levels between trimesters.Language and formatting
Proofreading should be done to correct minor errors in grammar and typos and add spaces where appropriate. Some of these are listed here:
Line 41: “2ndand 3rdtrimester” spaces should be added. Please correct this throughout the entire manuscript. Line 41: replace “to give” with “for”. Line 53: replace “was” with “has been”. Line 67: replace “because” with “based on”. Line 278: replace “about” to “approximately”, replace “of our study” with “in our study”. Line 284: replace “came” with “derived”. Line 323: replace “oppositely” with “conversely”. Line 318: replace “facts” with ”findings”. Line 323: replace “coworkers” with “colleagues”. Line 327: replace “adding…. to” with “combining… with” Line 356: “Based on our study findings argues” this should be rephrased.Author Response
RESPONSE TO REVIEWERS’ COMMENTS
Dear Editor:
We are pleased to resubmit the revised version for publication and truly thankful to you and to the reviewer for the deep and thorough review and proof reading. We have revised the present research paper in the light of the useful suggestions and comments. We hope our revision has improved the paper to a level of satisfaction. Our responses to your comments are given below in blue color as well as we mentioned the page number, paragraph number and line number.
With many thanks.
The authors
Reviewer 1:
Abstract
Line a3: a sentence is needed to describe the consequences of inadequate nutrition during pregnancy and the importance of assessing nutrition during pregnancy.
Response: It has been added.
Line 14: DRI abbreviation should be explained
Response: It has been explained.
Lines 21-22: the differences are between different women in the 3 trimesters, therefore is more suitable to write the sentence as follows: “Women in the 3rd trimester consumed significantly more protein….than women in the 1st and 2nd trimesters.”
Response: It has been amended.
Lines 22-23: the same as in comment 3.
Response: It has been amended.
Line 26: “Findings of the study showed that” is not needed.
Response: It has been deleted.
Line 28: replace “through” with “throughout”.
Response: It has been replaced.
Line 30: replace differences with increases.
Response: It has been replaced.
Line 32: “trimesters” instead “trimester”.
Response: It has been replaced.
Line 33: “exceeded” instead “exceeding”.
Response: It has been amended.
Line 33: “from the supplement not diet” should be rephrased. Much more focus should be given to the description of nutrients that were below or above the DRIs rather than the differences in intake between the trimesters. The conclusion of the abstract states this but it is not supported by the findings presented in the results section. The purpose and the meaning of comparing the levels of intake between trimesters is unclear. Which information do these comparisons provide in addition to the information whether the level is adequate or not in comparisons to the DRIs? Please add a sentence in the conclusions. The conclusions refer to the findings that some of the nutrients were below DRIs. However, other nutrients such as sodium exceeded DRIs. This also should be addressed here and in the discussion, including suggesting ways to resolve this problem.
Response: conclusion has been changed to: "While the intake of some nutrients from food alone remains below the DRIs in the diets of pregnant women, the intake of other nutrients are above the UL. Raising the awareness of pregnant women about their diet and how supplement intake can reduce the risk of inadequate intake for many micronutrients and improve their pregnancy outcomes is of great importance".
Introduction
Lines 53-59: the sentences here should be rephrased to state that even though the fertility rate in Jordan is relatively high there are still many problems affecting infant outcomes including malnutrition.
Response: It has been rephrased to: Even though the fertility rate in Jordan is relatively high there are still many problems affecting infant outcomes including malnutrition. However, Jordan is witnessing a decline in its high fertility rates [5]. Many factors in Jordan are affecting infant's morbidity and mortality rate including multigravida, mother malnutrition, fluctuations in financial capabilities, health care system, poverty, education and many other factors [6].
Methods
Line 98: “The FFQ was… validated…” “previously” should be added.
Response: It has been added.
Line 129: “previously” should be added.
Response: It has been added.
Lines 159-161: the cross-sectional design of the study in which different women are assessed in different trimesters is not suitable here because it doesn’t necessarily mean that all women showed increase in intake throughout pregnancy to the same extent. It would have been more appropriate to have a longitudinal study design, in which the same women assessed throughout all trimesters and use repeated measure ANOVA to analyse the within group differences across the 3 timepoints. Please address this and explain why you chose this design, including brief description of practical reasons if there were any.
Response: We selected the cross-sectional design to describe nutrients intake of Jordanian pregnant women at a specific point of time as well as to compare nutrients intake with DRI not to track the changes in nutrient intake across different trimesters. In addition, the study participants were recruited from a tertiary hospital and most of participants attained the antenatal clinics during their second and third trimesters. We enrolled only 50 pregnant women in first trimester because most of Jordanian pregnant women don't visit the physicians until they enter in the 2nd trimester. In addition, the risk of loss follow-up and panel attrition is high in longitudinal study. So, we could not carry out a longitudinal study design because we need a large sample size of pregnant women during first trimester which is not easy to enroll them from a tertiary hospital.
Results
Lines 216-221: this legend should be moved under table 3 and another one should be generated for table 2. P value should be explained that it is for the one-way ANOVA. Post-hoc test significance a and b should be explained.
Response: It has been added.
Line 220: replace “or” with “of”. Table 3 columns 2, 5, 8: it is unclear whether theses are the DRIs or the sample mean. It is better to split these into two columns for each trimester.
Response: It has been clarified.
Lines 231-232: this sentence belongs to the discussion and should be clarified. Line 246: legend for table 4 should be added, including an explanation of P values and post-hoc tests (a and b).
Response: It has been added.
Discussion
Line 282: add the name of the authors before the citation. Do the same throughout the discussion where appropriate.
Response: Sorry for these mistakes. All references have been added.
Lines 291-292: “… most pregnant women in our study took more folic acid in the first trimester with a significant decrease in the 2nd and 3rd trimesters” this sentence should be rephrased “women in the 1st trimester took more folic acid in comparison with women in the 2nd and 3rd trimesters.
Response: It has been replaced.
Line 292: a sentence should be added here that the level of folic acid exceeded the DRIs and the implications of this should be discussed, including whether there are negative consequences on pregnancy and infant outcomes. This is an example of a case where awareness to the importance of taking supplements exists but leads to over-consumption and therefore better education around this should be provided.
Response: This issue has been discussed and new study has been added.
Lines 290-307: the description should be either in relation to the comparison methods, i.e. describing all nutrients (or group of nutrients) that their levels found to be different across trimesters and followed by all nutrients compared to their DRIs, or describing for each nutrient the difference across trimesters and in comparison with DRIs. The way it is currently described is confusing and therefore hard to follow.
Response: It has been revised and changed in a way to make it easy to follow.
Line 290 and 297: the beginning of both sentences is very similar. It should refer from the beginning to the specific nutrient. The manuscript is mainly focused on the need to take supplements to correct deficiencies in some micronutrients. However, attention should also be given to ways to reduce energy and nutrients intake that exceeded DRIs, such as sodium, increase awareness to not overly consume supplements for nutrients that exceeded DRIs, such as folic acids, and encourage supplement consumption for nutrients that were below DRIs even after taking supplements. It will be good to have a few sentences describing the literature concerning awareness and education in Jordan around maternal nutrition during pregnancy.
Response: Unfortunately, we did not find any single study investigated the awareness and education in Jordan around maternal nutrition during pregnancy.
Line 350: add another limitation about the cross-sectional design of the study for comparing intake levels between trimesters.
Response: It has been added.
Language and formatting
Proofreading should be done to correct minor errors in grammar and typos and add spaces where appropriate. Some of these are listed here:
Line 41: “2ndand 3rdtrimester” spaces should be added. Please correct this throughout the entire manuscript. Line 41: replace “to give” with “for”. Line 53: replace “was” with “has been”. Line 67: replace “because” with “based on”. Line 278: replace “about” to “approximately”, replace “of our study” with “in our study”. Line 284: replace “came” with “derived”. Line 323: replace “oppositely” with “conversely”. Line 318: replace “facts” with ”findings”. Line 323: replace “coworkers” with “colleagues”. Line 327: replace “adding…. to” with “combining… with” Line 356: “Based on our study findings argues” this should be rephrased.
Response: All the above mentioned corrections have been performed.
The revised manuscript is attached and the corrections are declared in red color.

Reviewer 2 Report
Reema and colleagues present a cross-sectional study assessing nutrient intake among pregnant women in Jordan using meal frequency questionnaires, which showed that the intake of some nutrients from food alone was lower than DRI in the diet of pregnant women. The study is well structured and aids to a body of work, assessing the quality of food and nutrients in pregnant women and the possibilities of improving the nutritional value of the diet through the use of supplementation. Introduction and Discussion are correct, but the way in which results are presented needs improvement. However, I have a few comments and questions:
I would like to ask you to clarify whether the data analyzed in each trimester of pregnancy were consistent with normal and heterogeneous distribution. Considering that the consumption of some nutrients is not compatible with normal distribution.Did the authors determine the minimum sample size in order to ensure reliable results and the possibility of generalizing the obtained data
The value of the work would increase if the authors calculated the p-trend for the data presented in Table 4 and 6.
Tables 2 and 3 please remove the wording "percentage of energy from protein, fats ..." just say proteins (%) etc. 5.According to the reviewer Tables nr 5 and 7 could be combined and the significance of EAR and UL differences should be calculated for each trimester (where is applicable.
Author Response
Dear Editor:
We are pleased to resubmit the revised version for publication and truly thankful to you and to the reviewer for the deep and thorough review and proof reading. We have revised the present research paper in the light of the useful suggestions and comments. We hope our revision has improved the paper to a level of satisfaction. Our responses to your comments are given below in blue color as well as we mentioned the page number, paragraph number and line number.
With many thanks.
The authors
Reviewer 2:
Reema and colleagues present a cross-sectional study assessing nutrient intake among pregnant women in Jordan using meal frequency questionnaires, which showed that the intake of some nutrients from food alone was lower than DRI in the diet of pregnant women. The study is well structured and aids to a body of work, assessing the quality of food and nutrients in pregnant women and the possibilities of improving the nutritional value of the diet through the use of supplementation. Introduction and Discussion are correct, but the way in which results are presented needs improvement. However, I have a few comments and questions:
I would like to ask you to clarify whether the data analyzed in each trimester of pregnancy were consistent with normal and heterogeneous distribution. Considering that the consumption of some nutrients is not compatible with normal distribution.
Response: Yes, we considered the normal and heterogeneous distribution. We used Shapiro-Wilk test to assess normality of distribution of energy and nutrients intakes. When the variables were not normally distributed, data were log transformed before analysis.
Did the authors determine the minimum sample size in order to ensure reliable results and the possibility of generalizing the obtained data.
Response: Yes, we determined the minimum sample size for pregnant women in Amman. The estimated number of pregnant women is about 2000 women in 2017 at Amman. Accordingly, the calculated sample size was about 320 women. However, due to social and financial issues we were able to recruit just 300 women.
The value of the work would increase if the authors calculated the p-trend for the data presented in Table 4 and 6.
Response: Our statistician recommended just One Way ANOVA with LSD.
Tables 2 and 3 please remove the wording "percentage of energy from protein, fats ..." just say proteins (%) etc.
Response: It has been removed.
According to the reviewer Tables nr 5 and 7 could be combined and the significance of EAR and UL differences should be calculated for each trimester (where is applicable).
Response: combining the 2 tables will make it very crowded and not easy to read. And it is not applicable to calculate the significant differences of EAR and UL of each nutrients in each trimester. The EAR and/or UL have been not estimated for some nutrients such as niacin, folic acid, vitamin K, potassium, thiamine, riboflavin, and sodium. Moreover, the proportion of participants with UL intake was zero for most micronutrients except niacin, folic acid, magnesium, and sodium.
The revised manuscript is attached and the corrections are declared in red color.
